# Optimized Surgical Strategy for Adult Spinal Deformity: Quantitative Lordosis Correction versus Lordosis Morphology

**DOI:** 10.3390/jcm10091867

**Published:** 2021-04-26

**Authors:** Sang-Kyu Im, Ki Young Lee, Hae Seong Lim, Dong Uk Suh, Jung-Hee Lee

**Affiliations:** Department of Orthopedic Surgery, Graduate School, College of Medicine, Kyung Hee University, Seoul 02447, Korea; imskyu@naver.com (S.-K.I.); keyng39@hanmail.net (K.Y.L.); fofss86@daum.net (H.S.L.); sdu0810@hanmail.net (D.U.S.)

**Keywords:** lumbar degenerative kyphosis, optimal sagittal alignment, lordosis correction, lordosis morphology, lordosis distribution

## Abstract

Background: In surgical correction of adult spinal deformity (ASD), pelvic incidence (PI)-lumbar lordosis (LL) plays a key role to restore normal sagittal alignment. Recently, it has been found that postoperative lordosis morphology act as an important factor in preventing mechanical complications. However, there have been no studies on the effect of postoperative lordosis morphology on the restoration of sagittal alignment. The primary objective of this study was to evaluate the effect of postoperative lordosis morphology on achievement of optimal sagittal alignment. The secondary objective was to find out which radiographic or morphologic parameter affects sagittal alignment in surgical correction of ASD. Methods: 228 consecutive patients with lumbar degenerative kyphosis who underwent deformity correction and long-segment fixation from T10 to S1 with sacropelvic fixation and follow-up over 2 years were enrolled. Patients were divided according to whether optimal alignment was achieved (balanced group) or not (non-balanced group) at last follow-up. We analyzed the differences of postoperative radiographic parameters and morphologic parameters between two groups. Correlation analysis and stepwise multiple linear regression analysis was performed to predict the effect of PI-LL and morphologic parameters on the sagittal vertical axis (SVA). Results: Of 228 patients, 195 (85.5%) achieved optimal alignment at last follow-up. Two groups significantly differed in postoperative and last follow-up LL (*p* < 0.001 and *p* = 0.028, respectively) and postoperative and last follow-up PI-LL (*p* < 0.001 and *p* = 0.001, respectively). Morphologic parameters did not significantly differ between the two groups except lower lordosis arc angle (=postoperative sacral slope). In correlation analysis and stepwise multiple linear regression analysis, postoperative PI-LL was the only parameter which had significant association with last follow-up SVA (R^2^ = 0.134, *p* < 0.001). Morphologic parameters did not have any association with last follow-up SVA. Conclusions: When planning spine reconstruction surgery, although considering postoperative lordosis morphology is necessary, it is still very important considering proportional lordosis correction based on individual spinopelvic alignment (PI-LL) to achieve optimal sagittal alignment.

## 1. Introduction

The goal of surgical treatment for adult spinal deformity (ASD) is to restore sagittal alignment and achieve a solid arthrodesis in order to improve the patient’s quality of life [1]. Achieving optimal sagittal alignment is crucial to improve clinical symptoms and prevent complications, such as sagittal decompensation [1,2,3,4]. Appropriate lordosis correction is essential to that end; otherwise, decompensation may occur, and the consequential sagittal malalignment leads to the patient’s inability to maintain a stable posture in accordance with the “cone of economy” [5]. Many studies have reported the standard for lordosis correction, and currently, the Schwab formula, which quantitatively corrects the pelvic incidence (PI) and lumbar lordosis (LL) mismatch, is widely used. According to this formula, the PI-LL must be corrected within ±10° postoperatively in order to achieve optimal sagittal alignment and improve clinical symptoms, such as health-related quality of life [6,7].

To achieve such surgical goal, various surgical methods are used, including pedicle subtraction osteotomy (PSO), Smith–Petersen osteotomy, posterior column osteotomy (PCO), and posterior vertebral column resection [8,9,10]. Recently, various lordosis correction methods that use lateral lumbar interbody fusion (LLIF), which can further reduce complications, were introduced owing to advances in surgical instruments and techniques [11,12,13,14].

PSO allows a sharp angular postoperative LL, whereas LLIF allows a rounder shape [15]. Due to these differences, interest in lordosis morphology is increasing. Roussouly et al. [16,17] reported that the modeling of the sagittal profile is determined by the upper arc of lordosis, lower arc of lordosis, and lordosis apex. Further, lower lumbar distribution, a recently introduced qualitative parameter of lordosis correction, also affects lordosis morphology [18,19]. However, studies investigating the effect of postoperative lordosis morphology, as determined by these parameters, on restoration of sagittal alignment are still rare.

Therefore, the primary objective of this study was to evaluate the effect of postoperative lordosis morphology on achievement of optimal sagittal alignment. The secondary objective was to find out which radiographic or morphologic parameter affects sagittal alignment in surgical correction of ASD. For this purpose, among patients with ASD, we enrolled a single etiology of lumbar degenerative kyphosis (LDK), characterized by drop body syndrome (DBS) due to pure and severe sagittal malalignment [20,21], with a consistent level of long-segment fixation from T10 to S1 following deformity correction surgery.

## 2. Materials and Methods

### 2.1. Study Design

This study is a retrospective review of consecutive patients with ASD enrolled from 2005 to 2017.

The inclusion criteria were as follows: (1)Patients who had ASD accompanied by sagittal malalignment (sagittal vertical axis [SVA] > 50 mm, PI−LL > 10°, and pelvic tilt [PT] > 25°) with a minimum 2-year follow-up after deformity correction.(2)Patients who underwent deformity correction and long-segment fixation [22] from T10 to S1 with sacropelvic fixation as a surgical treatment by a single surgeon in a single institution.(3)A single etiology of LDK, redefined as DBS [20,21] patients who clearly showed atrophy of back musculature on magnetic resonance imaging as a diagnostic criterion for LDK and clinical signs, including walking difficulty with stooping, inability to lift heavy objects to the front, difficulty in climbing slopes, and need for elbow support when working in the kitchen, resulting in hard corns on the extensor surfaces [23,24,25].

Among patients who met the inclusion criteria, those who showed optimal sagittal alignment (SVA ≤ 50 mm) at the last follow-up were classified as the “balanced group”. Those who showed suboptimal sagittal alignment (SVA > 50 mm) at the last follow-up were classified as the “non-balanced group” [26].

### 2.2. Radiographic Measurements

Sagittal alignment was evaluated using lateral 14 × 36 inch full-spine radiographs obtained with the patients standing in a neutral, unsupported position with the fists-on-clavicle position [27]. All the digital radiographs were evaluated using validated software (Surgimap, Nemaris Inc., New York, NY, USA) [28].

We measured SVA, thoracic kyphosis angle (TK; T4–T10), LL angle (L1–S1), lower LL angle (LS; L4–S1), PI, PT, and sacral slope (SS) at the period of preoperative, postoperative, and at the last follow-up [29,30]. In order to evaluate the postoperative lordosis morphology, we measured the postoperative lordosis apex level (Apex) [31]. We measured the postoperative upper lordosis arc angle (UA) and lower lordosis arc angle (LA = SS) by dividing LL by the reference of the horizontal axis. As described by Roussouly et al. [16], LA is equal to SS. In addition, the postoperative lordosis distribution index (LDI; LS/LL × 100) [19] was measured, and we assessed lower LL using LA and LDI and total lordosis using LL. Further, the horizontal distance (ApS1) from the anterior-most margin of the apex vertebra to the posterior margin of the S1 endplate was measured (Figure 1) [32]. All postoperative morphologic parameters were measured two times by three professional orthopedic surgeons, and the mean measurements were used in analysis. To assess intra-observer and interobserver reliability, we calculated the interclass correlation coefficient (ICC). ICC 0.75 was set as the threshold for reliability [33]. Both the intra-observer and interobserver reliabilities were high (0.86–0.94 and 0.83–0.92, respectively).

### 2.3. Mechanical Complications

We evaluated the occurrence of mechanical complications such as proximal junctional kyphosis (PJK), rod fracture (RF) and number of revision cases of both groups. PJK was defined in accordance with the following criteria: proximal junction sagittal Cobb angle (inferior endplate of the upper instrumented vertebrae (UIV)-superior endplate of two vertebrae above the UIV) of ≥10°, with the change of angle at least 10° greater than the preoperative measurement [34]. Diagnosis of RF was based on the rod breakage with a recent fusion mass fracture seen on plain radiography and CT scans, confirmed by uptake on either bone scan or bone SPECT images.

### 2.4. Statistical Analysis

The Student’s *t*-test and Mann–Whitney test were used to evaluate the differences of radiographic parameters between the two groups. Chi-square test was used to evaluate the mechanical complications occurrence between the two groups. In addition, to evaluate the effect of PI-LL and postoperative lordosis morphologic parameters on postoperative SVA, the Pearson correlation test was used, and a stepwise multiple linear regression analysis was conducted to further assess the parameters that showed correlations. All statistical calculations were performed using SPSS software (version 25.0; IBM Corp., Armonk, NY, USA). A *p*-value < 0.05 was considered statistically significant.

## 3. Results

### 3.1. Baseline Characteristics

Table 1 presents the patients’ baseline characteristics. At the time of the study, the database included 310 patients, and after applying the inclusion criteria, 228 patients were identified for analysis. Patients’ average age at surgery was 71.5 years, and average length of follow-up was 45.3 months. All patients underwent eight segment fusions from T10 to S1. For sacropelvic fixation, 13 patients underwent conventional iliac screw fixation, and 215 patients underwent S2-alar-iliac screw fixation. For deformity correction and sagittal alignment restoration, 116 patients underwent PSO, and the accessory rod technique was performed with PSO to prevent pseudarthrosis, while 112 patients underwent multilevel LLIF with PCO [14].

### 3.2. Comparison of Radiographic Parameters and Mechanical Complications between the Balanced and Non-Balanced Groups

Table 2 presents the radiographic parameters and occurrence of mechanical complications of both groups. At the last follow-up, of 228 patients, 195 (85.5%) were in the balanced group with an SVA ≤ 50 mm and 33 were in the non-balanced group with an SVA > 50 mm (15.5%) (SVA: −1.5 mm vs. 70.2 mm, *p* < 0.001). Two groups significantly differed in postoperative and last follow-up LL (−68.7° vs. −61.1°, *p* < 0.001 and −64.3° vs. −57.6°, *p* = 0.028, respectively) and postoperative and last follow-up PI-LL (−13.2° vs. −3.5°, *p* < 0.001 and −8.8° vs. −0.4°, *p* = 0.001, respectively). Two groups also showed significant difference in postoperative and last follow-up PT (9.3° vs. 13.2°, *p* < 0.001 and 12.8° vs. 22.0°, *p* < 0.001). In the analysis of postoperative lordosis morphology, Apex, UA, LDI, and ApS1 did not significantly differ between the two groups, but LA (=postoperative SS) was significantly greater in the balanced group than in the non-balanced group (47.4° vs. 41.8°, *p* = 0.025). Mechanical complications occurred more in the last non-balanced group, but there were no significant differences between the two groups.

### 3.3. Relationship between the Radiographic Parameters and Last Follow-Up SVA

Table 3 shows correlations between the postoperative radiographic parameters and last follow-up SVA. Postoperative PI-LL and last follow-up PI-LL were significantly correlated with the last follow-up SVA (r = 0.338, *p* < 0.001 and r = 0.355, *p* < 0.001, respectively). Postoperative LL and last follow-up LL were also significantly correlated with the last follow-up SVA (r = 0.234, *p* < 0.001 and r = 0.307, *p* < 0.001, respectively). Among postoperative morphologic parameters, UA and LA, which were correlated with postoperative LL, were significantly correlated with the last follow-up SVA (r = −0.134, *p* < 0.05 and r = −0.171, *p* < 0.01, respectively). Apex, LDI, and ApS1 were not correlated with the last follow-up SVA.

In the stepwise multiple linear regression analysis, parameters showing multicollinearity were excluded, and the results showed that only postoperative PI-LL was significantly linearly associated with the last follow-up SVA (R^2^ = 0.134, *p* < 0.001). Other postoperative morphologic parameters were not associated with the last follow-up SVA (Table 4). Therefore, we confirmed that postoperative PI-LL affects sagittal alignment, whereas the morphologic parameters did not (Figure 2; Figure 3).

## 4. Discussion

Lordosis morphology can be determined using various radiographic parameters. Roussouly et al. [16,17] classified the sagittal alignment of young adults into four types using parameters such as the lordosis apex, UA, and LA (=SS). As the Roussouly sagittal profile type increases, the length and curvature of lordosis increases, and the apex becomes more proximal. Yilgor et al. [18,19] used LDI, the proportion of lower lordosis (=L4–S1) from total lordosis, from the Global Alignment and Proportion (GAP) score, which was designed for predicting mechanical complications after surgery for ASD. When LDI increases, the lower lumbar arc increases, and the lordosis apex becomes more caudal [35]. Moreover, ApS1 can be used to measure the horizontal distance from the apex to the LA [32]. These parameters can be used to determine lordosis morphology. To our knowledge, this study is the first to analyze the effect of postoperative lordosis morphology on optimal sagittal alignment using these parameters.

Among patients with ASD, we studied a single etiology known as LDK, which manifests as pure, severe sagittal malalignment. LDK frequently occurs in patients with an Oriental lifestyle who spend much of their time on the floor in a deep lumbar flexion posture. Yagi et al. [20] redefined LDK as DBS, as it is a distinct form of ASD, where patients show normal muscle strength and volume in the extremities but have significant local kyphosis with severe local back extensor muscle degeneration. They stated that DBS shows extreme primary sagittal-plane deformity with a high PI-LL mismatch [21]. Therefore, LDK requires greater sagittal correction in order to restore sagittal alignment. For this purpose, a large degree of lordosis correction and long-segment fixation are needed. Lee et al. [30] reported that overcorrection of LDK shows good radiologic and clinical outcomes and is effective in maintaining optimal sagittal alignment. In this study, we also observed significant differences in the postoperative and last follow-up LL and PI-LL between the balanced and non-balanced groups.

However, the two groups did not significantly differ in postoperative morphologic parameters, such as UA, LA, and LDI. Lordosis distribution, such as lower LL, greatly affects the distribution of weight loads [36]. Therefore, lordosis distribution has a great effect on the occurrence of mechanical complications, and Yilgor et al. suggested that appropriate LDI is conducive to prevent mechanical complications [18,19]. However, postoperative morphologic parameters did not affect sagittal alignment in our study. Hence, quantitative correction of PI-LL mismatch is more important than lordosis correction in consideration of morphology for restoring and maintaining sagittal alignment.

Postoperative morphologic parameters were not markedly associated with sagittal alignment according to the Pearson correlation test and multiple linear regression analysis of SVA. While UA and LA increased with increasing LL and thus were found to be correlated with the last follow-up SVA, multiple linear regression analysis revealed that only PI-LL significantly affected the last follow-up SVA. Lafage et al. [37] reported the correlations of the resection degree of PSO and PSO level with spinopelvic parameters. They reported that the resection degree of PSO is correlated with various spinopelvic parameters, such as TK, LL, SS, and correction degree of PT, but that the PSO level is only correlated with the correction degree of PT and not with other spinopelvic parameters. As the resection degree of PSO determines postoperative lordosis morphology, it also confirms that correction degree is more important than postoperative lordosis morphology for spinopelvic alignment, which is in line with our findings.

It is believed that the effect of the chain of correlation on the global sagittal alignment is the reason why correction degree plays a greater role than postoperative lordosis morphology in sagittal alignment. Among spinopelvic parameters, any segmental or regional change triggers adjacent segmental or regional change and ultimately alters the shape and position of the overall spinal curvature [38]. Berthonnaud et al. [39] reported that the sagittal plane comprises a linear chain linking system between adjacent anatomical segments from the head to the pelvis. According to the linear chain linking system, lumbar tilt, the tilt of the entire lumbar column, is correlated with thoracic tilt, which is the adjacent segment, and with spinopelvic parameters, such as PI, PT, SS, LL, and TK. Therefore, even in patients with surgically corrected lordosis, adequate correction, in consideration of the dynamic changes of the adjacent segments, is necessary to maintain optimal alignment rather than the morphology of the segments. Moreover, in this study, we confirmed that quantitative correction of PI-LL mismatch is a more effective criterion than the postoperative morphologic parameters for adequate lordosis correction.

In fact, in our study, the balanced group maintained optimal alignment until the last follow-up, with an average postoperative LDI of 39.5%, which is much lower than the ideal LDI of 50–80% according to the GAP score. Further, the postoperative apex was at the L2–L3 level, which is relatively more proximal compared to that in other studies [31,32]. Patients with LDK have a large PI [40], and the mean PI of the patients in our study was 55°. Overcorrection in these patients led to a greater lumbar curvature arc, which in turn made the apex more proximal and reduced the LDI below the ideal LDI. This is similar to the Roussouly sagittal profile type 4, which is characterized by a high PI, very curved and extended lordosis with lordosis apical vertebra at L3 or higher [16,17]. Further, Yilgor et al. studied patients with level four fusion or higher, while we studied patients who underwent long-segment fixation (eight levels) that included more proximal segments, which increased the percentage of UA and thereby contributed to such differences in the results. However, our previous study reported that the incidence of PJK is not different between under/overcorrection [41]. Therefore, after surgery, even when the LDI decreased and apex moved proximally due to overcorrection, the optimal alignment was still maintained well. Also, there was no difference in occurrence of mechanical complications between the two groups. However, further research is needed to determine the direct association between postoperative lordosis morphology and occurrence of mechanical complications.

This study has a couple of limitations. First, we could not analyze the differences in clinical outcomes according to postoperative lordosis morphology. Further, the incidences of other mechanical complications due to reduced LDI also need to be studied, and there is an ongoing study on this topic. Second, this study examined a single etiology known as LDK, so the patients had a large PI. Thus, in general, the apex was moved proximally, with a smaller LDI. Subsequent studies should examine postoperative lordosis morphology in patients with a small PI. Nevertheless, the strengths of this study are that this is the first study to examine the effect of postoperative lordosis morphology following surgical correction of ASD on optimal sagittal alignment, and that we studied a large patient population of more than 200 patients who underwent the same long-segment fixation for a homogeneous disease entity known as LDK. 

## 5. Conclusions

We showed that quantitative correction of PI-LL in consideration of individual spinopelvic alignment is more important than postoperative lordosis morphology as the criterion for the lordosis correction required to achieve optimal sagittal alignment. When planning spine reconstruction surgery for ASD, it is important to consider qualitative parameters in the prediction of mechanical complications. However, consideration of the proportional LL to pelvic incidence is still of utmost importance in order to achieve the surgical goals of maintaining optimal sagittal alignment and consequently, improving clinical symptoms.

## Figures and Tables

**Figure 1 jcm-10-01867-f001:**
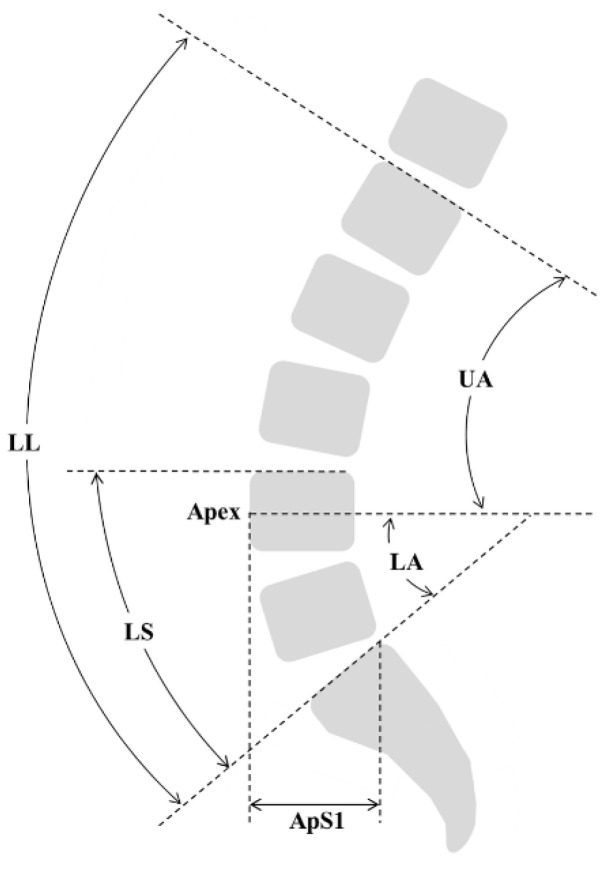
Parameters for evaluating postoperative lordosis morphology. LL, lumbar lordosis; LS, lumbosacral angle; UA; upper lordosis arc angle; LA, lower lordosis arc angle; Apex; lordosis apex, ApS1; horizontal distance from the anterior-most margin of the apex vertebra to the posterior margin of the S1 endplate.

**Figure 2 jcm-10-01867-f002:**
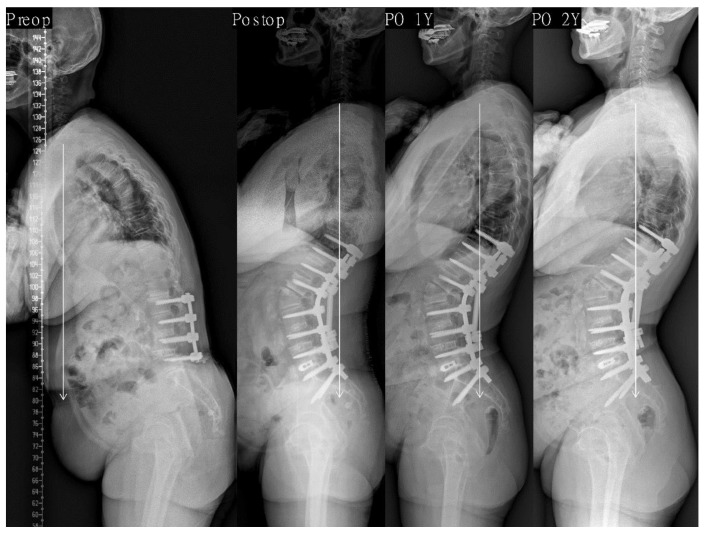
This 68-year-old female presented to us with degenerative sagittal imbalance with PLIF on L2-5 state (SVA + 202 mm, TK 9°, LL 15°, PI 45°, PT 20°, SS 25°). We performed PSO on L2 and PLIF on L5–S1. After surgery, lordosis apex was located at L2 and LDI was 25.0%. PI-LL was −18° (SVA −16 mm, TK 35°, LL −63°, PT 7°, SS 38°). Postoperative 1-year whole spine lateral radiograph showing normal sagittal alignment (SVA −29 mm) and is maintained optimal (SVA −23 mm) without occurrence of decompensation at the last follow-up period. PLIF indicates posterior lumbar interbody fusion; SVA, sagittal vertical axis; TK, thoracic kyphosis; LL, lumbar lordosis; PI, pelvic incidence; PT, pelvic tilt; SS, sacral slope; PSO, pedicle subtraction osteotomy; LDI, lumbar distribution index.

**Figure 3 jcm-10-01867-f003:**
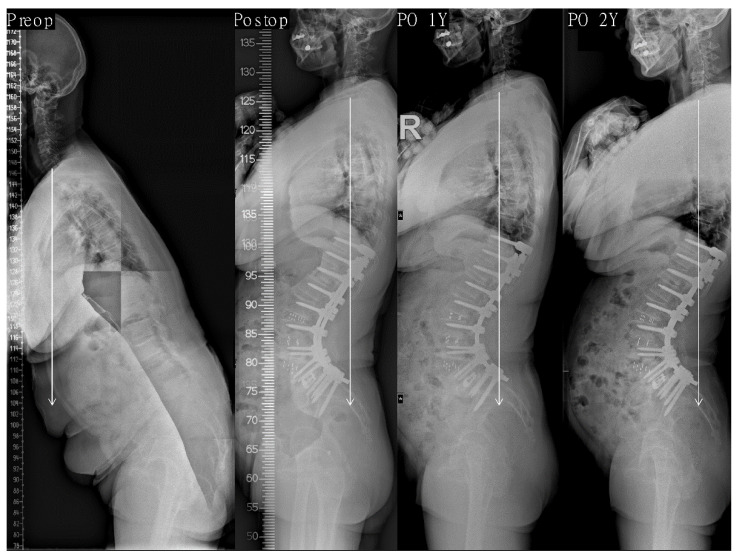
This 67-year-old female presented to us with degenerative sagittal imbalance (SVA + 251 mm, TK 20°, LL −20°, PI 56°, PT 35°, SS 21°). We performed PCO with LLIF on L2–5 and PLIF on L5–S1. After surgery, lordosis apex was located at L4 and LDI was 71.5%. PI-LL was −20° (SVA −46 mm, TK 39°, LL −76°, PT 1°, SS 55°). Postoperative 1-year whole spine lateral radiograph showing normal sagittal alignment (SVA −40 mm) and is maintained optimal (SVA −48 mm) without occurrence of decompensation at the last follow-up period. SVA indicates sagittal vertical axis; TK, thoracic kyphosis; LL, lumbar lordosis; PI, pelvic incidence; PT, pelvic tilt; SS, sacral slope; PCO, posterior column osteotomy; LLIF, lateral lumbar interbody fusion; PLIF, posterior lumbar interbody fusion; LDI, lumbar distribution index.

**Table 1 jcm-10-01867-t001:** Baseline characteristics (228 cases) ^†^.

Variables	
Age at surgery (years)	71.5 ± 5.4
Sex	
Female	222
Male	6
Surgical approach	
PCO with multilevel LLIF	112
PSO with accessory rod technique	116
UIV	T10
LIV	S1
Fused segments	8.0
Lumbosacral fusion	
PLIF	93
ALIF	118
None (previous fusion state)	17
Spinopelvic fixation	
Conventional iliac screw	13
S2AI screw	215
BMD (g/cm^2^)	0.991 ± 0.23
BMD T-score (g/cm^2^)	−1.223 ± 1.72
BMI (kg/cm^2^)	25.0 ± 1.9

^†^ Data are presented as mean ± standard deviation or number. PCO indicates posterior column osteotomy; LLIF, lateral lumbar interbody fusion; PSO, pedicle subtraction osteotomy; UIV, uppermost instrumented vertebra; LIV, lowermost instrumented vertebra; PLIF, posterior lumbar interbody fusion; ALIF, anterior lumbar interbody fusion; S2AI screw, sacrum 2 alar iliac screw; BMD, bone mineral density; BMI, body mass index.

**Table 2 jcm-10-01867-t002:** Comparison of radiographic parameters and mechanical complications between balanced and non-balanced groups ^†^.

Variables	Last Balanced Group (*n* = 195)	Last Non-Balanced Group (*n* = 33)	*p*-Value
SVA (mm)			
Preoperative	200 ± 65	193 ± 79	0.609
Postoperative	−14 ± 26	0 ± 34	0.062
Last f/u	−2 ± 27	70 ± 24	<0.001 *
Thoracic kyphosis (°) ^‡^			
Preoperative	6 ± 15	10 ± 13	0.399
Postoperative	27 ± 14	30 ± 12	0.414
Last f/u	35 ± 15	36 ± 16	0.815
Lumbar lordosis (°) ^‡^			
Preoperative	1 ± 20	0 ± 17	0.208
Postoperative	−69 ± 10	−61 ± 14	<0.001 *
Last f/u	−64 ± 11	−58 ± 18	0.028 *
Lumbosacral junction (°) ^‡^			
Preoperative	−6 ± 16	−9 ± 16	0.586
Postoperative	−28 ± 10	−26 ± 12	0.359
Last f/u	−28 ± 9	−27 ± 12	0.630
Pelvic incidence (°)	56 ± 11	58 ± 8	0.186
PI-LL (°)			
Preoperative	57 ± 20	55 ± 16	0.643
Postoperative	−13 ± 11	−4 ± 14	<0.001 *
Last f/u	−9 ± 10	0 ± 14	0.001 *
Pelvic tilt (°)			
Preoperative	30 ± 13	30 ± 16	0.907
Postoperative	9 ± 10	13 ± 9	<0.001 *
Last f/u	13 ± 10	22 ± 8	<0.001 *
Sacral slope (°)			
Preoperative	24 ± 13	28 ± 13	0.221
Postoperative	47 ± 9	41 ± 10	0.011 *
Last f/u	44 ± 9	42 ± 11	0.196
Postoperative lordosis morphologic parameters			
Upper lordosis arc (°)	−27 ± 7	−25 ± 9	0.357
Lower lordosis arc (= SS, °)	47 ± 9	41 ± 10	0.011 *
Apex (lumbar level)	15 ± 0	15 ± 0	0.978
Apex to S1 (mm)	65 ± 17	55 ± 19	0.072
LDI (%)	40 ± 12	44 ± 16	0.263
Mechanical complications			
PJK (%)	39 (20.0%)	11 (33.3%)	0.087
Rod fracture (%)	42 (21.5%)	9 (27.3%)	0.465
Revision surgery (%)	44 (22.6%)	9 (27.3%)	0.554

* Statistically significant (*p*-value < 0.05). ^†^ Data are presented as mean ± standard deviation, number (percentage), or number. ^‡^ Angles were measured as the Cobb angle. SVA indicates sagittal vertical axis; SS, sacral slope; LDI, lumbar distribution index; PJK, proximal junctional kyphosis.

**Table 3 jcm-10-01867-t003:** Correlation between postoperative lumbar sagittal parameters.

	Postop PI-LL	Last f/u PI-LL	Postop LL	Last f/u LL	UA	LA	Apex	ApS1	LDI
Last f/u SVA	0.338 **	0.355 **	0.234 **	0.307 **	−0.134 *	−0.171 **	−0.166	−0.076	−0.092

** Significant correlation was established at the 0.01 level. * Significant correlation was established at the 0.05 level. SVA indicates sagittal vertical axis; PI, pelvic incidence; LL, lumbar lordosis; UA, upper lordosis arc angle; LA, lower lordosis arc angle.

**Table 4 jcm-10-01867-t004:** Stepwise multiple regression analysis of influencing factors of last follow-up achievement of optimal alignment.

	Coefficient	*p*-Value	R^2^
Last f/u PI-LL (°)	0.997	<0.001 *	0.134

* Statistically significant (*p*-value < 0.05). PI indicates pelvic incidence; LL, lumbar lordosis.

## Data Availability

The data presented in this study are available on request from the corresponding author. The data are not publicly available, as participants of this study did not agree for their data to be shared publicly.

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
