# Peer review of "Optimized Surgical Strategy for Adult Spinal Deformity: Quantitative Lordosis Correction versus Lordosis Morphology"

_jcm, 2021, doi:10.3390/jcm10091867_

Round 1
Reviewer 1 Report
This manuscript is of interest for the spine deformity community. A lot of comments as depicted in the attached file make a revision necessary however.

Reviewer 2 Report
I would like to thank the authors for the opportunity to review their work. This series has the advantage of analysing a very homeogenic group among the adult deformity population.
The potential novelty here is looking at the importance of the lordosis shape as a better predictor of final sagittal alignment. However, the lower lordosis angle is equal to sacral slope, which has been extensively studied before.
There are some important missing data on the results sections. For the most important parameters (LDI and lordosis shape) we only have one-time point data. For the rest of the data we have preoperative, postoperative and final follow-up data. It is necessary to know how LDI and lordosis shape parameters were before and after surgery.
At the discussion, it is said that LDI was lower than the ideal proposed by GAP score, but from the study results it is unclear to me if this LDI value was before or after surgery.
In my view, the discussion fails to address the fact that LA=SS. SS=PI-PT and we know that PI is fixed, so changes in SS (and therefore LA) are probably due to an increase in PT due to recruitment of compensatory mechanisms.
I also think that limiting the study to correction parameters is in my view not enough to present this type of study. I’m aware that not everyone has the possibility to collect HRQoL, but the rate of mechanical complications in both groups should be given, as it is one of the main markers of successful treatment and is widely use in other publications, allowing for better comparison among different studies.
Round 2
Reviewer 2 Report
None
Author Response
.